# Long COVID and Hybrid Immunity among Children and Adolescents Post-Delta Variant Infection in Thailand

**DOI:** 10.3390/vaccines11050884

**Published:** 2023-04-23

**Authors:** Muttharat Jarupan, Watsamon Jantarabenjakul, Peera Jaruampornpan, Jarujan Subchartanan, Chayapa Phasomsap, Taweesak Sritammasiri, Sapphire Cartledge, Pintip Suchartlikitwong, Suvaporn Anugulruengkitt, Surinda Kawichai, Thanyawee Puthanakit

**Affiliations:** 1Department of Pediatrics, Faculty of Medicine, Chulalongkorn University, Bangkok 10330, Thailand; 2Center of Excellence for Pediatric Infectious Diseases and Vaccines, Department of Pediatrics, Chulalongkorn University, Bangkok 10330, Thailand; 3Thai Red Cross Emerging Infectious Diseases Clinical Center, King Chulalongkorn Memorial Hospital, Bangkok 10330, Thailand; 4Monoclonal Antibody Production and Application Research Team, National Center for Genetic Engineering and Biotechnology (BIOTEC), Pathum Thani 12120, Thailand; 5School of Medicine, University of Birmingham, Birmingham B15 2TT, UK; 6Department of Microbiology, Faculty of Medicine, Chulalongkorn University, Bangkok 10330, Thailand

**Keywords:** hybrid immunity, BNT162b2, Delta, Omicron, COVID-19, SARS-CoV-2 infection, neutralizing antibody titer, anti-SARS-CoV-2 IgG, post COVID-19 conditions, long COVID in children

## Abstract

This study aimed to assess long COVID, and describe immunogenicity against Omicron variants following BNT162b2 vaccination. A prospective cohort study was conducted among children (aged 5–11) and adolescents (aged 12–17) who had SARS-CoV-2 infection from July to December 2021 (Delta predominant period). Long COVID symptoms were assessed by questionnaires at 3 months after infection. Immunogenicity was evaluated by using a surrogate virus-neutralizing antibody test (sVNT) against the Omicron variant. We enrolled 97 children and 57 adolescents. At 3 months, 30 children (31%) and 34 adolescents (60%) reported at least one long COVID symptom, with respiratory symptoms prevailing (25% children and 32% adolescents). The median time from infection to vaccination was 3 months in adolescents and 7 months in children. At 1 month following vaccination, in children who received one-dose and two-dose BNT162b2 vaccines, the median (IQR) sVNT against Omicron was 86.2% inhibition (71.1–91.8) and 79.2% inhibition (61.5–88.9), respectively (*p* = 0.26). Among adolescents who received one-dose and two-dose BNT162b2 vaccines, the median (IQR) sVNT against Omicron was 64.4% inhibition (46.8–88.8) and 68.8% inhibition (65.0–91.2) (*p* = 0.64). Adolescents had a higher prevalence of long COVID than children. Immunogenicity against the Omicron variant after vaccination was high and did not vary between one or two doses of the vaccine in either children or adolescents.

## 1. Introduction

As of January 2023, more than 762 million severe acute respiratory syndrome coronavirus 2 (SARS-CoV-2) cases have been reported globally [1]. The reported symptoms of acute coronavirus disease 2019 (COVID-19) in children are less severe symptoms including fever, cough, sore throat, loss of taste and smell, malaise, fatigue, muscle pain, headache, shortness of breath, and also the reported no symptoms with fatality rate is less than 0.1%. However, additional research on the long-term consequences of SARS-CoV-2 infection in children and adolescents is needed [2]. Post-COVID-19 conditions or long COVID are defined as a wide range of new, returning, or ongoing health problems, present for four or more weeks, that develop during or after SARS-CoV-2 infection that cannot be explained by another diagnosis. These symptoms have a relapsing-remitting pattern and progressively worsen over time [3]. According to recent reports, some adults, adolescents, and children who contract SARS-CoV-2 have long-lasting symptoms and are unable to regain their prior levels of health. Some demographic groups may be more susceptible to post-COVID-19 conditions than others, including people who have experienced more severe COVID-19 illnesses, especially those who were hospitalized or needed intensive care, who had underlying health conditions prior to COVID-19, or who did not receive a COVID-19 vaccine [3]. The percentage of at least one persisting symptom is observed to be very high in children and adolescents. As reported in previous studies, the adolescent group even showed a higher post-COVID-19 symptoms percentage than the adults (45%) [4]. Adults reported fatigue, persisting dyspnea or difficulty breathing, and arthromyalgia as the most common post-COVID-19 symptoms. While in a systematic review and meta-analyses of post-COVID-19 conditions in children and adolescents in Europe, the most common symptoms described were neuropsychiatric alterations, e.g., sadness, tension, anger, depression, and anxiety; sleep disorders, e.g., insomnia, hypersomnia, and poor sleep quality; headache; fatigue; and respiratory symptoms. The risk of post-COVID-19 conditions increased in older age, females, more severe in acute COVID-19 illness, those with an underlying disease of obesity, allergic disease, or long-term health problems [2]. An additional study reported symptoms, on average 3–6 months following infection, of headache, sleep disturbance, fatigue, muscle and joint pain, respiratory problems, and palpitations [5]. However, most patients’ symptoms slowly improved with time [3].

Immunity after infection wanes rapidly in the first 2–3 months and then more slowly thereafter [6,7,8]; therefore, re-infection is not uncommon. In many, re-infection with COVID-19 occurs when new variants of concern (VOCs) escape immunity from the previous infection. SARS-CoV-2 VOCs, for example, the Omicron variant, required a higher level of neutralizing antibodies to prevent infection [9,10]. Immunization after the previous infection boosts immunity and protects against symptomatic infection [11].

The BNT162b2 mRNA vaccine has been approved as a two-dose regimen for adolescents aged 12–17 years (30 μg dose with a 3-week interval) [12] and for children aged 5–11 years (10 μg dose with an 8-week interval) [13]. Hybrid immunity is defined as an immune response in individuals who have had one or more doses of a COVID-19 vaccine and experienced at least one SARS-CoV-2 infection before or after the initiation of the vaccine. Hybrid immunity usually provides higher immunity than vaccination alone due to cell-mediated and humoral immunity induction, including memory B cells [10,14,15,16]. Several studies have found that immunity increases rapidly following one dose of the SARS-CoV-2 mRNA vaccine among individuals with previous SARS-CoV-2 infection. On the contrary, a second vaccination dose did not significantly increase the antibody titer [14,17,18]. Currently, the United States Centers for Diseases Control and Prevention (US-CDC) and Europe Centers for Diseases Control and Prevention (ECDC) have no special recommendations regarding vaccination for those infected previously with SARS-CoV-2. In clinical practice, children and adolescents who have been infected previously receive a 2-dose BNT 162b2 regimen, similar to the general population without previous SARS-CoV-2 infection [19].

This study aimed to describe post-COVID-19 conditions and residual symptoms three months after infection and assess the immunogenicity against Omicron variants of SARS-CoV-2 among children and adolescents who received BNT162b2 with either a one- or two-dose vaccination regimen following previous SARS-CoV-2 infection.

## 2. Materials and Methods

A prospective cohort study was conducted at the King Chulalongkorn Memorial Hospital, a tertiary care hospital in Bangkok, Thailand. Participants were eligible if (1) they were aged 5 to 17 years, and (2) they had a history of SARS-CoV-2 infection confirmed by nasopharyngeal swab positive for real-time reverse transcription polymerase chain reaction (RT-PCR) between July and December 2021 (Delta variant predominant period). The exclusion criteria were re-infection with SARS-CoV-2, which was determined by the presence of COVID-19 symptoms, and a positive RT-PCR again three months or more after the initial infection. Signed and written informed assent and consent were obtained from eligible participants and their guardians prior to study enrollment. This study was registered in the Thai Clinical Trials Registry (thaiclinicaltrials.org, TCTR20220510005) and was approved by the Institutional Review Board of the Faculty of Medicine, Chulalongkorn University (IRB No. 851/64).

Post-COVID-19 conditions questionnaires assessed 36 physical and mental health symptoms in the past four weeks. The questionnaire was self-reported by adolescents themselves, but completed by parents/caregivers for children aged 5–11 years old. Blood samples were collected at 3, 6, and 9 months following infection with SARS-CoV-2. Blood samples were centrifuged for serum separation and storage at −70 °C.

During the follow-up period of this study, the Thai Ministry of Public Health rolled out the BNT162b2 vaccination program. Starting in October 2021, BNT162b2 was available for adolescents as a 30 μg/dose with an interval of 3 weeks apart. Starting in February 2022, BNT162b2 was available for children aged 5–11 years, using a 10 μg/dose with an interval of 8 weeks apart. Participants voluntarily received either 1 or 2 doses of the BNT162b2 vaccine. In this study, we retrospectively selected blood samples from 2 to 6 weeks post-vaccination date to analyze SARS-CoV-2 immunogenicity. Immunogenicity was measured using anti-Spike-Receptor-Binding-Domain IgG antibody (anti-S-RBD IgG) and surrogate Virus Neutralization Test (sVNT) against Delta and Omicron variants.

### 2.1. Clinical Information of COVID-19 and Post- COVID-19 Conditions

Demographic and baseline clinical characteristics were obtained from a retrospective review of electronic medical records. The diagnosis of COVID-19 and the evaluation of disease severity were based on diagnostic criteria of the Department of medical services, Ministry of Public Health, Thailand, in 2021. SARS-CoV-2 infection was diagnosed based on the detection of viral RNA in respiratory tract swab samples by RT-PCR. Clinical severity was categorized as asymptomatic, non-pneumonia (defined as having fever and/or upper respiratory tract symptoms, sore throat, loss of taste, loss of smell, malaise, fatigue, muscle pain, headache, gastrointestinal symptoms such as nausea, vomiting, diarrhea, rash), and pneumonia (dyspnea, crepitation sound, desaturation in the resting state, or chest imaging showing infiltration).

We used post-COVID-19 conditions questionnaires at 3, 6, and 9 months following infection developed by the Royal College of Pediatricians of Thailand (Appendix A) to assess residual or new symptoms after recovery from acute SARS-CoV-2 infection, as shown in Figure 1. The questionnaires included 36 symptoms of physical and mental health divided into six organ systems as follows: (1) pulmonary system: cough, anosmia, rhinorrhea/nasal congestion, chest pain or chest tightness, difficulty breathing, and dyspnea; (2) dermatologic system: hair loss, and rash; (3) gastrointestinal system: diarrhea, constipation, abdominal pain, and difficulty swallowing; (4) neurological system: headache, insomnia, somnolence, dizziness, syncope, abnormal movement, tremor, numbness, loss of concentration, communication skill problems, and seizure; (5) musculoskeletal system: muscle pain, abnormal posture, and arthralgia; and (6) other symptoms: fever, fatigue, sore throat, loss of appetite, loss of taste, weight loss, urinary problems, irregular menstruation, conjunctivitis, and blurred vision. In addition, questionnaires included the date of BNT162b2 vaccination(s) and the date of recurrent SARS-CoV-2 infection (if any).

### 2.2. SARS-CoV-2 Immunogenicity and Laboratory Assays

This study measured anti-Spike-Receptor-Binding-Domain IgG antibody (anti-S-RBD IgG) against Wuhan-Hu-1 using an enzyme-linked immunosorbent assay (ELISA) and neutralizing antibodies against Delta and Omicron variants (surrogate Virus Neutralization Test; sVNT). The recombinant spike RBD proteins were derived from B.1.617.2 or B.1.1.529 (sublineage BA.1) for sVNT assay. Anti-S-RBD IgG level was reported in binding-antibody units (BAU/mL) after the conversion of OD450 values using the World Health Organization (WHO) International Standard curve and presented as the geometric mean titer (GMT). In addition, the sVNT was performed and reported the result as % inhibition. All tests were developed in-house by the National Center for Genetic Engineering and Biotechnology (BIOTEC) during 2020 to 2022 [20,21].

### 2.3. Data analysis and Statistics

Descriptive analysis was used for baseline and clinical characteristics. Continuous variables were reported as a median and interquartile range (IQR). Categorical variables were reported as numbers with frequencies and percentages. Anti-S-RBD IgG was reported as the geometric mean titer with 95% CI. sVNT was reported as a median (IQR). Unpaired t-tests and median tests were used as statistical tests. This study used the anti-S-RBD IgG level above 506 BAU/mL as the lower limit for protective antibody levels, aligning with the WHO recommendation [22]. As a threshold that correlates with protective efficacy, the proportion of sVNT ≥ 68% inhibition was chosen [23]. Statistical significance was defined as a two-sided *p*-value < 0.05. Stata version 13.0 (Stata Corp., College Station, TX, USA) was used for analysis.

## 3. Results

### 3.1. Baseline Characteristics

From November 2021 to June 2022, 154 participants (97 children and 57 adolescents) were enrolled (Figure 1). The demographic data are shown in Table 1. The median age was 9 (IQR 7, 13) years; 95 participants (61%) were female, and 20 participants (13%) were diagnosed with COVID-19 pneumonia. One hundred twenty-one participants (79%) were diagnosed with non-pneumonia, which presented with mild symptoms, including only a fever and/or upper respiratory tract symptoms, sore throat, loss of taste, loss of smell, malaise, muscle pain, headache, rash, gastrointestinal symptoms such as nausea, vomiting, and diarrhea. Thirteen participants (8%) were diagnosed as asymptomatic. Most of the participants were healthy with no previous diseases; only 3% reported allergic rhinitis, 2% asthma or chronic lung disease, and 2% obesity. Fifty-seven children and 37 adolescents received either one or two doses of BNT162b2. During follow-up, 18 participants, 14 (9%) children and 4 (3%) adolescents, had SARS-CoV-2 re-infection with a median (IQR) of 6.6 (4.7–7.8) months from the first infection. All 14 children who were re-infected did not receive any BNT162b2 vaccination. However, all four adolescents received BNT162b2 vaccination after SARS-CoV-2 infection; three received one dose, and one with two doses. Re-infection among four adolescents who received the vaccine(s) occurred within a range of 2.5–3.5 months after the last vaccine.

### 3.2. Post-COVID-19 Conditions or Long COVID

Among 154 participants, 64 (42%) reported persisting symptoms three months after infection. A higher percentage of adolescents had persistent symptoms following infection compared to children. The data are shown in Figure 2 and Figure 3. Overall, 14 (14%) children and 12 (21%) adolescents had one persisting symptom. Seven (7%) and four (3%) children had two and three persisting symptoms, respectively, while five (9%) and five (9%) adolescents had two and three persisting symptoms, respectively. The most commonly reported symptoms following SARS-CoV-2 infection were respiratory symptoms, including dyspnea, difficulty breathing, rhinorrhea, and cough. Adolescents also reported hair loss, muscle pain, and fatigue as common persisting symptoms.

Six months after SARS-CoV-2 infection, 5% and 1% of children had one or two persisting symptoms, respectively. Meanwhile, 15% and 4% of adolescents reported one and two persisting symptoms, respectively. However, 4% of adolescents had three persisting symptoms. Nine months after SARS-CoV-2 infection, 2.5% of children reported having sleep disturbance. Whereas 12% of adolescents still had one persisting symptom, 12% had three persisting symptoms. Common symptoms reported by adolescents nine months after infection were hair loss, respiratory symptoms (rhinorrhea, dyspnea, difficulty breathing), sleep problems (insomnia, somnolence), and headache.

### 3.3. Immunogenicity against SARS-CoV-2 Delta and Omicron Variants

#### 3.3.1. Children Aged 5–11 Years

Data on immunogenicity were available for 97 children at three months after infection and 87 children at six months after infection; the geometric mean titer of anti-S-RBD IgG was 113 (95%CI 90, 142) and 150 (95%CI 116, 194) BAU/mL at 3 and 6 months after infection, respectively. The median of % inhibition for sVNT against the Delta variant was 28.9% (IQR 14.7, 55.6) and 22.6% (IQR 12.8, 45.2) at 3 and 6 months after infection, respectively. In contrast, 0% inhibition for sVNT against the Omicron variant was observed at 3 and 6 months.

According to vaccine availability, some children received the first dose of 10 μg of BNT162b2 at a median time of approximately seven months after infection. Thirty-eight children received one dose of BNT162b2 and had immunogenicity measurement one month after vaccination: the geometric mean titer of anti-S-RBD IgG was 1669 (95%CI 1394, 1998) BAU/mL, the median of % inhibition for sVNT against Delta was 99.8% (IQR 99.7, 99.9), and against Omicron was 86.2% (IQR 71.1, 91.8). Nineteen children received two doses of BNT162b2. One month after the last vaccination, the geometric mean titer of anti-S-RBD IgG was 1808 (95%CI 1540, 2121) BAU/mL. %inhibition for sVNT against Delta was 99.9% (IQR 99.8, 100.0), and against Omicron was 79.2% (IQR 61.5, 88.9) (Table 2).

#### 3.3.2. Adolescents Aged 12–17 Years

Three months after Delta-variant SARS-CoV-2 infection, the geometric mean titer of anti-S-RBD IgG was 116 (95% CI 68, 198) BAU/mL, and the median of % inhibition for sVNT against Delta and Omicron was 41.2 (IQR 16.7–80.8) and 0%, respectively.

However, one month after a booster with the BNT162b2 vaccine in 24 adolescents, the geometric mean titer of anti-S-RBD IgG was 3010 (95% CI 2208, 4105) BAU/mL. The median of % inhibition for sVNT against Delta and Omicron was 99.8% (IQR 99.6, 100.0) and 64.4% (IQR 46.8, 88.8), respectively. Among 13 adolescents who received two doses one month after the last vaccination, the geometric mean titer of anti-S-RBD IgG was 3498 (95% CI 3071, 3944) BAU/mL, and the median of % inhibition for sVNT against Delta and Omicron was 99.9% (IQR 99.8, 100.0), and 68.8% (IQR 65.0, 91.2), respectively (Table 3).

## 4. Discussion

In this study, post-COVID-19 conditions are common in young people. Sixty percent of adolescents and 31% of children reported at least one symptom at three months post SARS-CoV-2 infection. The common symptoms among adolescents were pulmonary symptoms, hair loss, muscle pain, and fatigue. In contrast, the parents of younger children reported rhinorrhea, cough, and difficulty breathing as common symptoms among children. Three months post-infection, we found low neutralizing antibodies against the SARS-CoV-2 Delta variant, and there were no measurable neutralizing antibodies against the SARS-CoV-2 Omicron variant. However, after receiving 1 or 2 doses of BNT162b2 vaccines, the neutralizing antibody titers increased to >99 % inhibition for the Delta variant and 66.4–86.2% inhibition for the Omicron variant. Interestingly, there was no difference in the antibody titer among individuals who received either one or two doses of BNT162b2.

The risk of post-COVID-19 conditions or long COVID in adolescents was found to be higher than in children. These data are consistent with prior studies demonstrating that adolescents had a better expression of symptoms in self-reported questionnaires of post-COVID-19 conditions [2,24]. We found respiratory symptoms were the most commonly reported symptom in children with post-COVID-19 conditions. At the same time, adolescents were more likely to report headaches or fatigue, similar to what has been reported in the adult literature [25]. One explanation could be that adolescents self-reported symptoms, whereas symptoms in children were obtained from their parents/caregivers, which might bias towards more visible symptoms such as cough. In addition, the other possible explanations are a strong innate immune response, reduced expression of angiotensin-converting enzyme-2 (ACE2) receptors, and active thymic function in young children leading to a milder severity of acute COVID-19 and long COVID. Post-COVID-19 conditions in this study population, especially those who had struggled with respiratory symptoms, were not associated with comorbidities (asthma, chronic lung diseases, or allergic diseases). Additionally, this study discovered reports of persistent symptoms following infection, even though children often had asymptomatic or mild symptoms when infected with the Delta variant of SARS-CoV-2. The lack of COVID-19 vaccination in our study population of children and adolescents prior to SARS-CoV-2 infection may be the root cause of the high prevalence of post-COVID-19 conditions. In a previous study from Denmark, post-COVID-19 conditions resolved in 54–75% of both children and adolescents within 1–5 months [26]. In this study, only 6% and 2.5% of children still suffered long-haul COVID at six months and nine months, respectively. In comparison with adolescents, one-quarter consistently experienced post COVID-19 symptoms at nine months. The management of children with post-COVID-19 is based on professional judgment and guidelines from adults. An integrated approach tailored to the individual child’s presentation and symptoms is the goal with appropriate workup while avoiding the harms of overtesting. The children and adolescents in our study were reassured to supportive care and received mental support, which seemed to improve over time.

This report analyzed the immunogenicity while the Delta variant was predominant from 3 to 9 months after primary SARS-CoV-2 infection. The anti-S-RBD IgG and sVNT against Delta variant response in the remaining unvaccinated population found the immunity waning in 3 months, similar to other studies [6,7,8]. The sVNT against the Omicron variant in the unvaccinated group at 3- and 6-months follow-up is very low, suggesting infection with the Delta variant does not provide cross-protection against infection with the Omicron variant in these unvaccinated individuals. The Omicron variant has a large number of mutations in the spike protein of the virus, which is the target of the vaccines made against it. Therefore, some of these mutations may reduce the ability of the vaccine-induced antibodies to recognize and neutralize the virus [27].

Prior work indicates that a stronger humoral response to SARS-CoV-2 infection (anti-S-RBD IgG and neutralizing antibodies) correlates with protection from re-infection and reduces severe disease, hospitalization, and death [22,23]. Results from our study indicate that SARS-CoV-2 immunity might be transiently protective after infection; in this cohort, 9% of children reported re-infection with a median of 6.6 months after the first infection and no serious illness upon re-infection. These data are consistent with data from US children from March 2020 to June 2022, showing that previous infection with the Delta variant had 50–70% effectiveness in preventing Omicron re-infection, but still maintained protection from the severe disease [28]. In addition, SARS-CoV-2 vaccines have been reported to broaden the diversity of T-cell immunity against different SARS-CoV-2 spike proteins following infection, thus enhancing protection against variants and inducing long-lasting memory B-cell populations [11,14].

Our study found that vaccination with either 1 or 2 doses of BNT162b2 after natural infection led to increased antibody titers in both children and adolescents. A study of hybrid immunity [14] found that the response to the second vaccine dose did not significantly increase antibody titers in previously infected individuals [11,17]. The second dose of BNT162b2 was given at only 3–4 weeks intervals, which is unlike the second booster dose that American Academy of Pediatrics (AAP) recommended for high-risk individuals should be at an interval of least 4–6 months for booster effect. In our study, children had a higher level of sVNT against the Omicron variant than adolescents, which could be explained by a prolonged interval period between the first infection and vaccine administration (8 months vs. 4 months). This study showed that a longer period between infection and a booster vaccination increases the quality of memory B cells and enhances abilities to increase the immune response [14]. Therefore, consideration could be given to delaying the interval between infection and vaccination by 3–6 months. However, the timing of vaccination depends on individual factors such as the risk of severe diseases, COVID-19 spreading status in the community, and changes in the SARS-CoV-2 strain. In addition, those with hybrid immunity compared with three doses of mRNA vaccine in those with no previous infection had similar surrogate neutralizing antibody levels [29].

The strengths of this study include the design being a longitudinal cohort study of children and adolescents following COVID-19 infection, allowing us to not only characterize post COVID-19 conditions in Thai children, but also describe the humoral immunity of individuals pre- and post-BNT162b2 vaccination following prior natural infection. This type of study is important to inform clinical decision-making as the predominant circulating strains of SARS-CoV-2 continue to change. However, we lack information on the immunological response to one or two doses of the BNT162b2 vaccination in the same individual. The limitation is that this study had a small number of adolescents included in the post COVID-19-conditions assessment and measured antibody responses at only 1 month post-vaccination without data on long-term immunity and vaccine effectiveness.

## 5. Conclusions

Nearly half of children and adolescents reported at least one long COVID symptom at three months after SARS-CoV-2 infection. In comparison to children, adolescents had a larger percentage of post-COVID-19 conditions. At three months after infection, those of the study population who had previously contracted SARS-CoV-2 and had post-COVID-19 symptoms had not been administered the COVID-19 vaccine prior to infection. The majority of the persisting symptoms being reported in adolescents were similar to adults. The prevalence of symptoms decreased over time. Natural immunity rapidly fell three months following SARS-CoV-2 infection; however, the immune response was substantially increased after the BNT162b2 vaccine booster administration. In children and adolescents previously infected with SARS-CoV-2, neutralizing antibody levels one month following vaccination were similar regardless of whether one or two doses were received.

## Figures and Tables

**Figure 1 vaccines-11-00884-f001:**
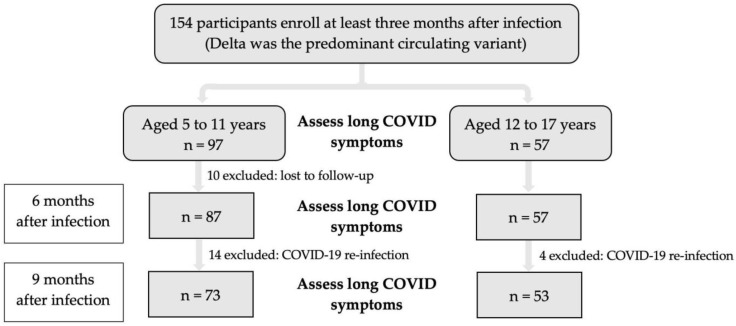
Diagram of enrollment.

**Figure 2 vaccines-11-00884-f002:**
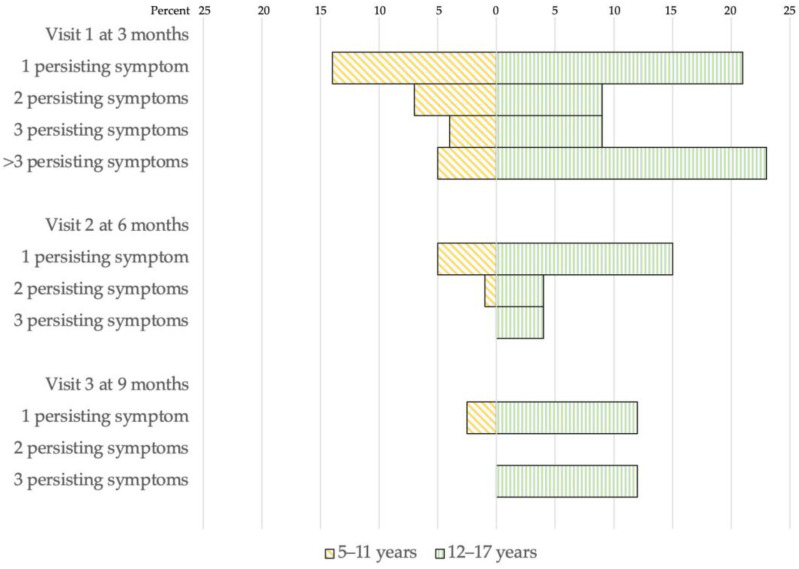
Prevalence of post-COVID-19 conditions at 3, 6, and 9 months after infection among 97 children and 57 adolescents.

**Figure 3 vaccines-11-00884-f003:**
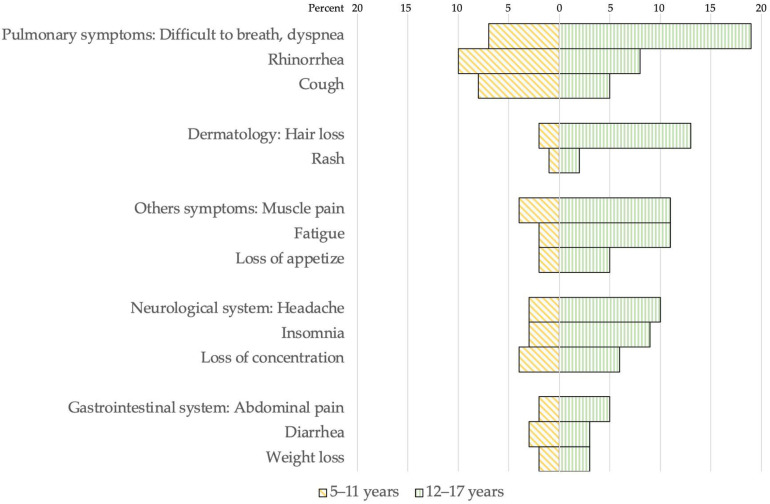
Persisting symptoms of post-COVID-19 conditions at 3 months after infection among 97 children and 57 adolescents.

**Table 1 vaccines-11-00884-t001:** Baseline characteristics from the first enrollment of study participants.

Characteristics	All n = 154	Children 5–11 Years Old n = 97	Adolescent 12–17 Years Old n = 57
Age, median (IQR), year	9 (7, 13)	8 (6, 9)	13 (12, 14)
Gender, female, n (%)	95 (61)	59 (60)	35 (61)
Height, median (IQR), cm	139 (125, 156)	128 (120, 138)	159 (153, 164)
Bodyweight, median (IQR), kg	38 (25, 49)	27 (22, 40)	51 (44, 68)
Severity of COVID-19, n (%)			
- Non-Pneumonia	134 (87)	85 (88)	49 (86)
- Pneumonia	20 (13)	12 (12)	8 (14)
Underlying disease, n (%)			
- Asthma, chronic lung disease	3 (2)	1 (1)	2 (3)
- Allergic rhinitis	4 (3)	1 (1)	3 (5)
- Obesity	3 (2)	2 (2)	1 (2)

cm, centimeter; kg, kilogram.

**Table 2 vaccines-11-00884-t002:** Immunogenicity following Delta variant SAS-CoV-2 infection in children aged 5–11 years.

	No Vaccine n = 87 ^1^	1 Dose of BNT162b2 n = 38	2 Doses of BNT162b2 n = 19	*p*-Value
Interval from vaccination to blood drawn, days	NA	32 (27, 35)	31 (28, 37)	0.56 ^2^
Time from infection to blood drawn, months	6.1 (5.9, 6.3)	8.5 (7.1, 8.8)	8.6 (8.1, 9.5)	
The Geometric mean titer of anti-S-RBD IgG against wild-type, BAU/mL, mean (95%CI)	150 (116, 194)	1669 (1394, 1998)	1808 (1540, 2121)	
sVNT against Delta variant, % inhibition	22.6 (12.8, 45.2)	99.8 (99.7, 99.9)	99.9 (99.8, 100.0)	
sVNT against Omicron variant, % inhibition	0	86.2 (71.1, 91.8)	79.2 (61.5, 88.9)	0.26 ^3^

Data except for the Geometric mean titer of anti-S-RBD IgG were shown as median (interquartile range). ^1^ Only 87 children had a blood draw at visit 2 because ten children were lost to follow-up at visit 2. ^2^ Unpaired *t*-test of the interval from the last vaccine dose to the blood draw. ^3^ Unpaired *t*-test of % sVNT against Omicron variant between 1 and 2 vaccine doses.

**Table 3 vaccines-11-00884-t003:** Immunogenicity following Delta variant SAS-CoV-2 infection in adolescents aged 12–17 years.

	No Vaccine n = 16	1 Dose of BNT162b2 n = 24	2 Doses of BNT162b2 n = 13	*p*-Value
Interval from vaccination to blood drawn, days	NA	25 (17, 35)	39 (15, 52)	0.64 ^1^
Time from infection to blood drawn, months	3.3 (2.5, 4.2)	3.9 (3.3, 4.2)	3.5 (3.0, 3.7)	
The Geometric mean titer of anti-S-RBD IgG against wild-type, BAU/mL, mean (95%CI)	116 (68, 198)	3010 (2208, 4105)	3498 (3071, 3944)	
sVNT against Delta variant, % inhibition	41.2 (16.7–80.8)	99.8 (99.6, 100.0)	99.9 (99.8, 100.0)	
sVNT against Omicron variant, % inhibition	0	64.4 (46.8, 88.8)	68.8 (65.0, 91.2)	0.64 ^2^

Data except for the geometric mean titer of anti-S-RBD IgG were shown as median (interquartile range). ^1^ Unpaired *t*-test of interval from last vaccine dose to blood draw. ^2^ Unpaired *t*-test of % sVNT against Omicron variant between 1 and 2 vaccine doses.

## Data Availability

The data supporting this study’s findings are available from the corresponding author upon reasonable request.

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
