# Peer review of "Long COVID and Hybrid Immunity among Children and Adolescents Post-Delta Variant Infection in Thailand"

_vaccines, 2023, doi:10.3390/vaccines11050884_

Round 1

Reviewer 1 Report

This manuscript by Jarupan and colleagues describes reported long COVID symptoms in Thai children and adolescents at 3, 6, and 9 months following symptomatic COVID-19 during the time of Delta predominance, and also reports immunogenicity following mRNA vaccination in these same children. The data are interesting and the study is generally well conducted. However, a number of issues dampen my enthusiasm for this work as presented. First, it would appear that the entire section of the discussion pertaining to the long COVID results are missing, making it impossible for me to fully assess the merits of this manuscript. Second, while both the long COVID and immunogenicity aspects of the study each appear to be well-conducted, I fear that these are two quite distinct topics and I suspect that each aspect would be better served as individual papers, not forced together into a single manuscript. Finally, I believe that the paper suffers from a relatively poorly presented materials and methods section, which makes following the paper quite challenging. Greater detail and clarity regarding the timing of the assessments, how and when certain participants were excluded, and a more specific CONSORT diagram would significant improve the readability. Specific comments, including attempts at improving/refining the written English where applicable, are detailed below.

Abstract

·       It is unclear in the abstract when the vaccinations were given and the time frame for assessing immunogenicity.

·       Line 26-7: shouldn’t this say “surrogate virus-neutralizing antibody test?.” Also, the units of % inhibition are not clear.

·       Line 34: suggest “was high” and “did not vary”

Introduction

·       Lines 43-45: the precise definition of long COVID is unsettled and the term “post-COVID conditions” has generally superseded the use of “long COVID.” Of note, the cited reference for the listed definition is no longer available. Consider rephrasing “long COVID” here and elsewhere to “post-COVID-19 conditions” and citing updated references from CDC or WHO.

·       Lines 55-56: suggest rephrasing “Immunity after infection wanes rapidly in the first 2-3 months then more slowly thereafter…”

·       Line 63: suggest rephrasing the definition of hybrid immunity here. It cannot be hybrid if the immune response is following vaccination but prior to infection. The intent is understood but the wording is imprecise.

·       Lines 65-66: suggest rewording to “due to induction of both cell mediated and humoral immunity, including memory B cells.”

·       Line 73: should be “similar to”

Materials and Methods

·       Lines 84-85: The exclusion criterion as written are unclear. Does this mean that a child who had re-infection within 2 months was eligible but one with re-infection at 4 months was excluded? I suspect this is meant to state that children with re-infection within 3 months of initial infection were exclude, but that is not how this is written.

·       Line 94: to clarify, was blood centrifuged for serum (plasma) separation and storage?

·       Line 96: suggest “During the follow-up period”

·       Line 100: was there a national recommendation for 2 doses? Under what circumstances would a participant not receive the second dose? For blood samples, was post-dose 2 blood selected for all subjects with 2 doses? Or based on timing, could some participants have had post-dose 1 blood sampled despite having gone on to receive the second dose?

·       Line 102: suggest “for analysis of SARS-COV-2…”

·       I would suggest presenting here the timeline for questionnaire administration—it is not presented in the methods anywhere and makes the results very confusing at first glance

·       Line 118: suggest “difficulty breathing”

·       Line 120: suggest “difficulty swallowing” or dysphagia

·       Line 120: suggest “somnolence” vs “sleepy”

·       Line 122: abnormal movement should be clarified—this could be neurologic as well?

·       Line 124: suggest “loss of taste”

·       Line 125: remove parenthesis

·       Line 130: this refers to recombinant spike generated using the relevant sequences, not actual viral protection, correct?

·       Line 133: missing period

·       Line 141: would specify the median tests: Mann-Whitney?

·       Line 145: specify one-sided or two-sided p-value

Results

·       Line 150: remove period after enrolled

·       Line 152: reword “37 adolescents WHO received”

·       Line 153: no hyphen needed after 1 or 2 doses

·       Line 157: suggest “three received one dose and one received two doses.”

·       Line 158: Who are the children who were re-infected 2.5-3.5 months after the last vaccine?

·       Figure 1: First box, would reword “154 participants enrolled at least three months after infection”

·       Are the 14 children and 4 adolescents identified here as having been excluded the same as those referenced in the text? If so, it should be stated that in the text they were excluded from analysis following the initial assessment. If some were infected greater than 6 months following initial infection, I would revise Figure 1 to specify the exact time intervals when the children were re-infected and excluded from subsequent analysis. For example, at least some children were re-infected greater than 6 months after initial infection, so shouldn’t their exclusion have occurred between the 6 and 9 month assessment?

·       Table 1: the percentages for female sex are incorrect

·       Figure 2: suggest listing the total N of both children and adolescents. Would consider including similar figures for the 6 and 9 months assessments, if not in the main body then as supplemental data.

·       Line 189: the fact that antibody titers were higher at 6 months rather than 3 months seems awfully suspicious to me. No vaccination has occurred in the interval?

·       Line 192: suggest rephrasing to “respectively, whereas 0% inhibition for sVNT agains the Omicron variant was observed at 3 and 6 months.”

·       Line 194: should be “According to vaccine availability…”

·       Line 196: suggest rewording to “…and had immunogenicity measurement 1 month after vaccination: geometric mean titer of…”

·       Line 198:  suggest “and against Omicron…”

·       Line 199: remove “And” and change to “2 doses”

·       Line 200: suggest “the geometric mean titer…”

·       Line 201: suggest “against Omicron…”

·       Line 203: should be “Adolescents”

·       How come there are no antibody data for adolescents 6 months post-infection? Is this because most of them started getting vaccinated before that time point?

·       Line 210: suggest “who received 2 doses”

·       Table 2: suggest “1 dose” and “2 doses”

·       Table 2: footnotes 2 and 3 should be “unpaired t-test.” Suggest rephrasing: “Unpaired t-test of interval from last vaccine dose to blood draw” and “unpaired t-test of % sVNT against Omicron variant between 1 and 2 vaccine doses”

·       Table 3, title should be “adolescents.” Same comments regarding footnotes as above.

Discussion

·       There is no discussion provided regarding long COVID. Was a chunk of the discussion inadvertently deleted?

·       It should be noted that very few older adolescents were included in the long COVID assessment

·       Lines 237-239 are unclearly worded

·       Line 242: Which study is this referring to?

·       Lines 249-250: suggest “but also to describe humoral immunity of individuals pre- and post-BNT162b2 vaccination following prior natural infection.”

·       Lines 252: while RCTs are the standard, I highly doubt that new RCTs specifically testing this hypothesis will be feasible or practical to conduct given the ubiquity of infection at this point

·       Line 255: this sentence is unclearly worded

Reviewer 2 Report

The manuscript entitled “Long COVID and hybrid immunity among children and adolescents post-Delta variant infection in Thailand” shows very important results of immunity after covid vaccination and long covid symptoms in adolescent and children.

I have few minor concerns.

Abstract

There is a need to add the major long covid symptoms (name and percentage) observed in children and adolescents during the study in the abstract.

Introduction

Add a paragraph, about the symptoms, which are observed normally during the covid infection.

Percentage of at least one covid symptoms is observed very high in children and adolescent group. Adolescent group even showed higher post covid symptom percentage than the adults (45) as reported in previous studies (page 1, line 46, and reference 4).

Introduction, line 41. Children typically have mild symptoms, whereas your studies showed atleast 1 long covid symptom in high number among children and adolescent.

Line 106. Demographic and baseline clinical characteristics were obtained from retrospective review of electronic medical records.

From the baseline clinical data. What are the major symptoms observed in the study participants during the infection? Out of these major symptoms, which covid symptoms is dominantly observed in children and adolescent after 3 and 6 months, is missing in the study.

Figure 1: Improve text font and writing style, to make figure 1 more clear.

Discussion is short. There is need to add more references and to discuss your results with already published data.

Questionnaire used in the study, may be added as supplementary material.

Reviewer 3 Report

Overall, It's a good study, and its well presented. no negative feedback

Reviewer 4 Report

In the manuscript ‘Long COVID and hybrid immunity among children and adolescents post-Delta variant infection in Thailand’ the authors compared long COVID symptoms and immunogenicity among two groups pre and post BNT162b2 vaccination. The manuscript is written in a sound scientific language, experiments and data analysis performed accurately.

I have the following suggestions to the authors for the revision of the manuscript.

1.       Discussion: Line 218- There seem to be a few sentences missing at the start of the discussion, as the discussion starts mid sentence.

2.       Line 63-64: The authors should rewrite or redefine the meaning of Hybrid immunity something on the lines of ‘ hybrid immunity is immunity acquired after having a full series of vaccination and also had a prior infection in any order’

3.       Line 130: The spike what ? antibody ?

4.       Line 244: replace has with have

5.       Line 255-257: Reframe this last sentence.  

6.       Line 262 -263: …..after administration with a booster with BNT162b2 vaccine booster.

Round 2

Reviewer 1 Report

The authors have nicely responded to virtually all prior comments. With the inclusion of the discussion regarding post-COVID symptoms and the revisions, this manuscript is now suitable for publication. I have only a few minor editing suggestions, as detailed below. One comment pertains to accuracy however—Table 1 still lists incorrect percentages for female gender (explained below).

Abstract:

Line 22: I would clarify that immunogenicity was measured in children who had received vaccine following prior infection.

Line 25: “%inhibition of” is probably not needed here as the addition of “test” in line 26 and addition of “inhibition” following the percentages later makes this clearer now.

Line 30: would add “in” in front of “children”

Methods:

Line 109: suggest “for serum separation and storage”

Table 1: The percentages for female sex are still incorrect. For children 5-11, 59/97=60%, not 38% as reported, and for adolescents 12-17, 35/57=61%, not 22% as reported.

Discussion:

Line 265: Data are plural, suggest “These data are consistent…”

Line 305: AAP should be defined at first use as American Academy of Pediatrics, and would reword to “…for high-risk individuals should be at an interval of least 4-6 months for booster effect.”
